# Virus specificity and nucleoporin requirements for MX2 activity are affected by GTPase function and capsid-CypA interactions

**Bailey Layish[1], Ram Goli[1], Haley Flick[1], Szu-Wei Huang[2], Robert Z. Zhang[1], Mamuka Kvaratskhelia[2], Melissa Kane[1]***

**1** Department of Pediatrics, Division of Infectious Diseases, University of Pittsburgh School of Medicine, Pittsburgh, Pennsylvania, United States of America, **2** Division of Infectious Diseases, University of Colorado Anschutz Medical Campus, Aurora, Colorado, United States of America

\* kaneme@pitt.edu

## Abstract

Human myxovirus resistance 2 (MX2/MXB) is an interferon-induced GTPase that inhibits human immunodeficiency virus-1 (HIV-1) infection by preventing nuclear import of the viral preintegration complex. The HIV-1 capsid (CA) is the major viral determinant for sensitivity to MX2, and complex interactions between MX2, CA, nucleoporins (Nups), cyclophilin A (CypA), and other cellular proteins influence the outcome of viral infection. To explore the interactions between MX2, the viral CA, and CypA, we utilized a CRISPR-Cas9/AAV approach to generate CypA knock-out cell lines as well as cells that express CypA from its endogenous locus, but with specific point mutations that would abrogate CA binding but should not affect enzymatic activity or cellular function. We found that infection of CypA knock-out and point mutant cell lines with wild-type HIV-1 and CA mutants recapitulated the phenotypes observed upon cyclosporine A (CsA) addition, indicating that effects of CsA treatment are the direct result of blocking CA-CypA interactions and are therefore independent from potential interactions between CypA and MX2 or other cellular proteins. Notably, abrogation of GTP hydrolysis by MX2 conferred enhanced antiviral activity when CA-CypA interactions were abolished, and this effect was not mediated by the CA-binding residues in the GTPase domain, or by phosphorylation of MX2 at position T151. We additionally found that elimination of GTPase activity also altered the Nup requirements for MX2 activity. Our data demonstrate that the antiviral activity of MX2 is affected by CypA-CA interactions in a virus-specific and GTPase activity-dependent manner. These findings further highlight the importance of the GTPase domain of MX2 in regulation of substrate specificity and interaction with nucleocytoplasmic trafficking pathways.

## Author summary

HIV-1 entry into the nucleus is an essential step in viral replication that involves complex interactions between the viral capsid and multiple cellular proteins, including the proline isomerase cyclophilin A. Nuclear entry of HIV-1 and other primate lentiviruses is

**Data Availability Statement:** Raw values used to generate graphs are deposited in the BioStudies database (https://www.ebi.ac.uk/biostudies/) under

the accession number S-BSST1357 (doi: 10.6019/S-BSST1357) Project number: NAHDAP-198575.

**Funding:** o This work was supported by NIH - National Institute of Allergy and Infectious Diseases (R01AI162172 to MKa) and (R01 AI162665 to MKv) and by Gilead Sciences Research Scholars Award to MKa. The content is solely the responsibility of the authors and does not necessarily represent the official views of the National Institutes of Health or of Gilead Sciences. The funders had no role in study design, data collection and analysis, decision to publish, or preparation of the manuscript.

**Competing interests:** The authors have declared that no competing interests exist.

inhibited by the antiviral protein MX2. Here, we show that direct interactions between capsid and cyclophilin A affect the antiviral activity and specificity of MX2, and that these interactions are altered when the enzymatic activity of MX2 is eliminated. We demonstrate that abolishing enzymatic activity of MX2 also alters the requirements for nuclear pore complex components for viral restriction. Our study provides new insights into how the enzymatic function of MX2 affects inhibition of lentiviral nuclear import.

## Introduction

Innate immune detection of viral infection results in the production of type I interferons (IFNs), which in turn induce the expression of hundreds of IFN-stimulated genes (ISGs) that confer an 'antiviral' state, impeding the propagation of many viruses [1,2]. Among the antiviral ISGs are the dynamin-like guanosine triphosphatases (GTPases) *MX1* (*MXA*) and *MX2* (*MXB*). Both proteins contain an amino-terminal GTPase domain connected to a carboxy-terminal stalk domain via a tripartite bundle signaling element (BSE) [3]. MX1 is exclusively found in the cytoplasm, while MX2 contains a nuclear localization signal (NLS)-like sequence in its first 25 amino acids and localizes at the nuclear pore complex (NPC) [4–6]. MX1 has long been known as a broadly acting and potent antiviral protein [3], while MX2 was ascribed with antiviral function in 2013 as an inhibitor of HIV-1 and other primate lentiviruses, and later shown to also restrict replication of herpesviruses and hepatitis B and C viruses [6–11].

MX2 inhibits HIV-1 infection prior to the chromosomal integration of proviral DNA, but after the completion of reverse transcription [6,7,12], and current models suggest that it acts by preventing nuclear import of the viral preintegration complex. The viral capsid (CA) is the major viral determinant of MX2 sensitivity, and several single amino-acid substitutions in CA have been identified that confer partial or complete resistance to MX2 [6–8,13,14]. MX2 has also been found to directly bind the HIV-1 CA, however the relevance of this binding for viral inhibition is unclear since MX2-resistant CA proteins are still efficiently bound by MX2 [15,16].

The antiviral activity of MX1 has been extensively investigated, and current understanding indicates that engagement of MX1 with components of viral replication complexes (such as the nucleoprotein of orthomyxoviruses) results in higher order oligomerization, followed by GTP hydrolysis and conformational changes which lead to the perturbation of viral transcription/replication via mislocalization of targeted components, blocking nuclear translocation, and/or disruption of the functional integrity of the replication complex. GTPase function and higher order oligomerization are generally required for antiviral activity of MX1 [3]. Conversely, GTP binding and hydrolysis are dispensable for the antiviral activity of MX2 [6,7,17], and although dimerization is essential, higher order oligomerization is not required for viral restriction [16,18]. Finally, while anti-viral specificity of MX1 is largely determined by a disordered loop (L4) in the stalk domain [19], antiviral specificity of MX2 is determined by the N-terminal domain (NTD) [13,17], indicating that the mechanisms underlying inhibition of viral infection by MX1 and MX2 are distinct. The antiviral activity of MX2 is also regulated by protein phosphorylation. Phosphorylation of serines 14, 17, and 18 in the NTD suppresses antiviral activity, reduces interactions with the viral CA and accumulation at the nuclear envelope, while dephosphorylation by myosin light chain phosphatase (MLCP) upon IFN stimulation restores antiviral function [20]. Phosphorylation status of additional residues throughout all domains of MX2 also alter the antiviral activity and specificity of MX2, with some phosphomimetic mutants exhibiting increased antiviral activity and the capacity to inhibit MX2-resistant

HIV-1 CA mutants [21]. Thus, a variety of determinants throughout MX2 modulate its function and viral specificity.

Sensitivity of HIV-1 to MX2 activity is affected by complex interactions between CA and cellular proteins involved in the early stages of HIV-1 infection, including nucleoporins (Nups) and the peptidyl-prolyl isomerase cyclophilin A (CypA) (reviewed in [22]). CypA-CA interactions can affect many early steps of HIV-1 replication, disruption of the interaction between HIV-1 and CypA, either genetically or by cyclosporine A (CsA) addition, reduces the efficiency of HIV-1 infection (reviewed in [23]), and some studies have indicated that CypA-CA interactions influence nuclear entry [24,25], and more recently CypA has been shown to protect the HIV-1 CA from restriction by TRIM5alpha; [26,27]. However, it remains unclear whether the isomerase activity of CypA plays a role in its pro-viral function, since mutations that abolish enzymatic activity also affect CA-binding [28]. Indeed, defining a precise role for CypA in HIV-1 infection has proved difficult because CsA treatment has diverse enhancing and inhibiting effects on infection that are determined by CA mutations and target cell type, suggesting that the function of CypA in HIV-1 infection is differentially affected by other cellular proteins in various contexts (reviewed in [23,29]).

Here, we investigated the effects of CA-CypA interactions on the antiviral activity and specificity of MX2. We find that the effects of CypA on MX2 activity against HIV-1 are the direct result of CA-CypA interactions. We also identified cell-type-dependent determinants for the GTPase and oligomerization requirements for MX2 activity. We show that abrogation of GTP hydrolysis by MX2 conferred enhanced antiviral activity when CA-CypA interactions were abolished. This activity is independent of previously reported CA-binding residues in the GTPase-domain of MX2. Finally, we show that GTPase-deficient MX2 has altered nucleoporin requirements for antiviral activity. These results indicate that GTPase function alters substrate specificity and antiviral function of MX2.

## Results

### The effects of CypA on MX2 activity are mediated by direct interactions between CypA and the viral CA

Multiple previous reports have demonstrated that CsA addition abolishes the antiviral activity of MX2 against wild-type HIV-1 (HIV-1$_{WT}$) [8,30,31], and CA mutants unable to bind CypA (eg. G89V and P90A) are MX2-resistant [6–8,12,14,15,31] suggesting that MX2 is only active against CypA bound HIV-1. However, CA mutations have multiple pleiotropic effects, and CsA treatment targets multiple cyclophilin proteins [23,32,33]. Furthermore, CypA also has multiple cellular functions that affect protein folding, trafficking, interaction, and activation [34]. Therefore, to address whether there are capsid-independent effects of CypA on HIV-1 infection or MX2 activity, we generated CypA knock-out cell lines as well as cells expressing CypA from its endogenous locus but with specific point mutations introduced to alter CA-binding specificity. TRIMCyp from both rhesus and pigtailed macaques does not restrict HIV-1 infection and differ from human CypA at only two amino acid positions [D66N and R69H], both of which are outside the active site (Fig 1A) and differently affect binding to lentiviral capsids [35,36]. We first generated cells ectopically expressing HA-tagged owl monkey TRIMCyp (omkTRIMCyp), or chimeric proteins in which the CypA domain was replaced with human CypA (TRIM-huCypA) or human CypA with D66N, R69H, or both D66N/R69H substitutions. We then challenged these cells with HIV-1$_{WT}$, its derivatives encoding CA mutations with altered MX2-sensitivity, and other lentiviruses (S1 Fig). Cells were also treated with the CypA inhibitor cyclosporine A (CsA) at the time of infection to abolish CypA:CA interactions. As expected, HIV-1$_{WT}$ infection was potently inhibited (~200-fold) by TRIM-huCypA$_{R69H}$ in

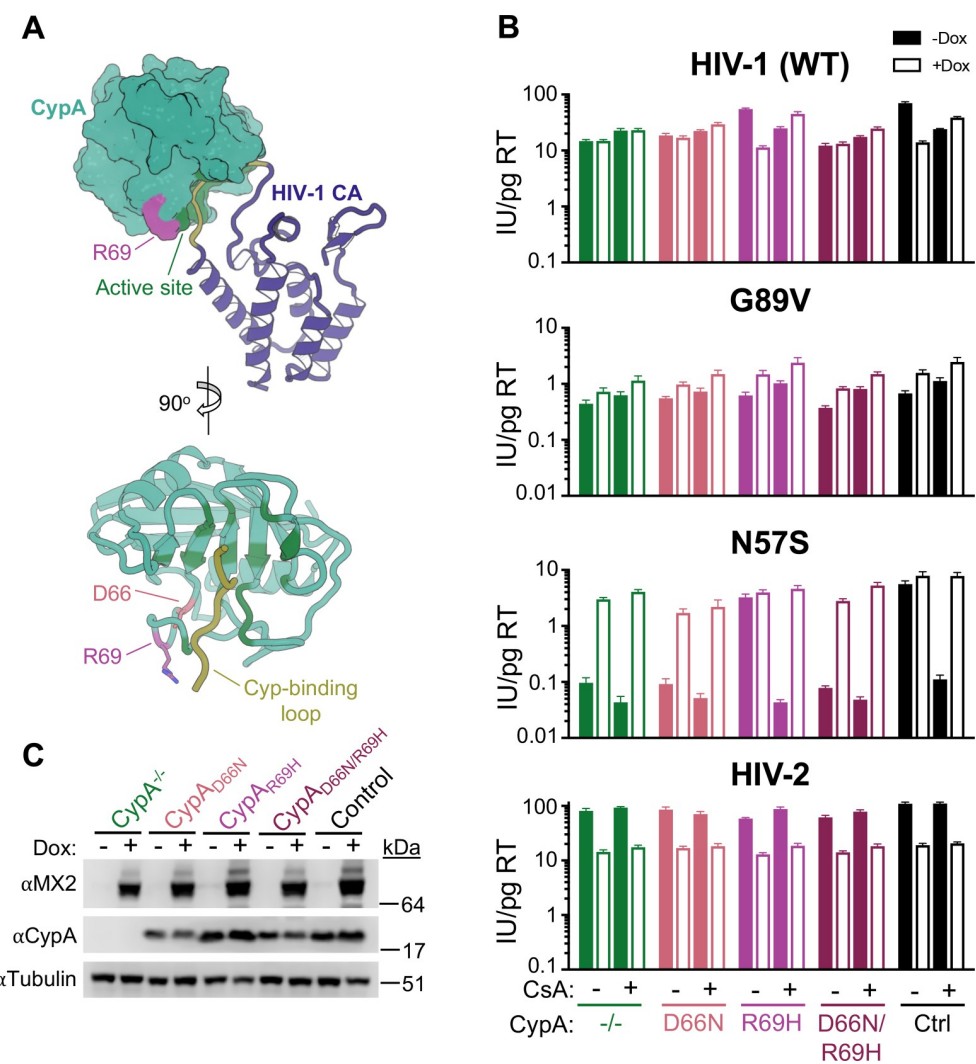

**Fig 1. The effects of CsA on HIV-1 sensitivity to MX2 are the direct result of blocking CA-CypA interactions.** A) Structure of an HIV-1 capsid monomer in complex with CypA (PDB 1AK4). Critical active -site residues that interact with CA are indicated in dark green, amino acids outside the active site that affect capsid recognition highlighted in pink/magenta. The cyclophilin-binding loop of HIV-1 CA indicated in yellow. B) Infection of control or CypA-mutant HT1080 cells expressing doxycycline-inducible MX2 in the presence (open bars) or absence (filled bars) of doxycycline (Dox) and present or absence of CsA with the indicated GFP reporter viruses. Titers are represented as mean + sem of infectious units (IU) per pg of reverse transcriptase (RT), n≥6 technical replicates combined from two-three independent experiments. Statistical analysis in S1 File. C) Western blot analysis of doxycycline-inducible MX2, CypA, and tubulin loading control.

a manner reversible by CsA treatment, while TRIM-huCypA$_{D66N}$ and TRIM-huCypA$_{D66N/R69H}$ were inactive against HIV-1$_{WT}$. With the exception of the cyclophilin-binding loop mutant G89V, all HIV-1 CA mutants tested exhibited similar sensitivity to the chimeric TRIM-huCypAs as the WT virus. HIV-2 on the other hand, was inhibited (~10-fold) by TRIM-huCypA$_{D66N}$ and TRIM-huCypA$_{D66N/R69H}$, but not TRIM-huCypA$_{R69H}$, while SIVmac was insensitive to all TRIMCyps since it does not bind cyclophilins (S1 Fig) [25,37,38].

To generate CypA mutant cell lines, we employed a CRISPR/Cas9-AAV-based gene engineering approach to target the *PPIA* locus (S2A Fig). CypA knock-out cells were generated by introduction of guide RNAs targeting intron 1 and exon 4, resulting in either a deletion of most of the coding sequence or a frameshift mutation that abrogated expression (S2B Fig). To

generate point mutant cells, an AAV containing donor-DNA for homology-directed repair was introduced with D66N and/or R69H mutations (S2A and S2C Fig). Single cell clones with the desired mutation were confirmed by sequencing of both genomic DNA and mRNA. We then tested the ability of CypA in control and point mutant clones to bind HIV-1 CA *in vitro* (S3 Fig), and found that both $CypA_{WT}$ and $CypA_{R69H}$, but not $CypA_{D66N}$ or $CypA_{D66N/R69H}$ pelleted with WT or N57A mutant CA, while CA(G89V) did not bind to CypA from any of the cell lines.

CypA mutant cell lines were then transduced with a lentiviral vector for doxycycline-inducible expression of MX2 and infected with GFP-reporter viruses in the presence or absence of CsA (Fig 1). CsA addition in HT1080 cells is known to reduce infection of HIV-1$_{WT}$ and abolish the antiviral activity of MX2 ([30], Figs 1B, S2D, and S5). We observed similarly reduced levels of infectivity and a lack of MX2 sensitivity in the absence of CsA treatment in CypA$^{-/-}$, $CypA_{D66N}$, and $CypA_{D66N/R69H}$, but not $CypA_{R69H}$ cells. Importantly, we did not observe clone-dependent effects on HIV-1 infection or MX2 activity (S2D Fig). Mutation or deletion of CypA had no effect on infection of the HIV-1$_{G89V CA}$, or on the slight enhancement of infection by this mutant upon MX2 expression. We previously reported a dramatic sensitivity of the cell-cycle-dependent and MX2-resistant HIV-1$_{N57S CA}$ to CsA addition that is reversed by MX2 [30]. Here we observed the same ~50-fold reduction in infectivity and rescue by MX2 in CsA-treated and CypA$^{-/-}$, $CypA_{D66N}$, or $CypA_{D66N/R69H}$, but not $CypA_{R69H}$ cells (Figs 1B and S5). Since the N57A CA mutation was utilized for *in vitro* CA binding experiments, we also confirmed that HIV-1$_{N57S CA}$ exhibited the same phenotype, as did the N57D mutation, although this mutant had lower overall levels of infectivity which were slightly enhanced by MX2 even in the absence of CsA treatment (S4 and S5 Figs). Infectivity and MX2 sensitivity of HIV-1$_{A92E CA}$, HIV-1$_{N74D CA}$, and HIV-1$_{T210K CA}$ in CypA$^{-/-}$, $CypA_{D66N}$, or $CypA_{D66N/R69H}$ cells also phenocopied CsA treatment in WT and $CypA_{R69H}$ cells. Furthermore, no additional effects of CsA treatment on HIV-1$_{WT}$ or CA mutant infection were observed in CypA knockout or mutant cell lines (Figs 1B, S4, and S5). Collectively, these data demonstrate that the effect of CsA on HIV-1 infection and MX2 sensitivity are the direct result of abrogating CA-CypA interactions and do not involve other cyclophilins, and further suggests that they do not involve other cellular functions of CypA. However, since it is unknown whether the conformational changes in CypA resulting from substitution at residue 66 affect interactions between CypA and any of its cellular substrates, we cannot definitively exclude the possibility that $CypA_{D66N}$ fails to isomerize a cellular substrate relevant for these interactions.

In contrast to HIV-1, HIV-2 sensitivity to MX2 is unaffected by CsA treatment ([30], Figs 1B and S5), although HIV-2 CA does interact with CypA with low affinity [35]. The CypA D66N and R69H mutations increase binding affinity, making HIV-2 sensitive to rhesus and pigtailed macaque TRIMCyps ([35,36,38] and S1 Fig). On the other hand, huCypA interaction with SIVagmTAN do not appear to involve CypA residues 66 or 69, as this virus was restricted by all TRIMCyp fusions tested (although this virus is not restricted by owl monkey TRIMCyp with the NH mutation, highlighting the complexity of lentiviral CA-CypA interactions [36]). However, both HIV-2 and SIVagmTAN infection and MX2 sensitivity were unaffected in CypA-knockout and mutant cells, similar to SIVmac, which does not bind cyclophilins, (Figs 1B, S4, and S5). These results indicate that unlike restriction of HIV-1, inhibition of HIV-2, SIVmac, and SIVagmTAN by MX2 is independent of CA-CypA interactions.

## Determinants for CA-specific effects of MX2 on lentivirus infection

We next sought to identify the determinants in MX2 that affect the CA- and CypA-specific sensitivity of lentiviruses to MX2 activity. We expressed a panel of (C-terminally myc tagged)

MX2 mutants that altered the N-terminal CA-interacting domain, GTP binding and hydrolysis, or oligomerization (Fig 2) in HeLa and HT1080 cells and first determined their effect on HIV-1$_{WT}$ infection in the presence and absence of CsA. Murine leukemia virus (MLV) was included as an MX2-insensitive negative control. Inhibition of HIV-1$_{WT}$ infection by chimeric MX1 expressing the N-terminal 91 amino acids of MX2 (MX1$_{MX2-NTD}$) was reversed by CsA treatment in both HeLa and HT1080 cells (even enhancing HIV-1$_{WT}$ infection in CsA-treated HT1080 cells to a greater degree than MX2). As expected [6,7,17,21,39], MX2 with a deletion of the N-terminal 25 amino acids (ΔN25), mutation of the N-terminal triple-arginine motif (RRR$_{11-13}$AAA), or phosphomimetic S14D/S17D/S18D (SSS$_{14,17,18}$DDD) did not inhibit HIV-1$_{WT}$ infection, while non-phosphorylatable S14A/S17A/S18A (SSS$_{14,17,18}$AAA) had antiviral activity similar to WT MX2, that was abrogated in the presence of CsA in HT1080, but not HeLa cells (Figs 3, S6, and S7). Both the monomeric M574D and limited oligomer forming R455D mutants exhibited no anti-HIV-1 activity, while dimeric YRGK$_{487-490}$AAAA had extremely reduced antiviral activity in both HeLa and HT1080 cells. This is in contrast with previous reports that have shown full antiviral activity of dimeric MX2 in U87-MG and HOS cells [16,18] and suggests that the requirements for MX2 oligomerization are cell-type specific. The T151A mutant of MX2, which binds, but does not hydrolyze GTP [5], restricted HIV-1 infection in both HeLa and HT1080 cells, confirming that GTPase activity is dispensable for anti-HIV-1 activity [6,7]. The T151A mutant also restricted HIV-1$_{WT}$ infection in the presence of CsA in both cell lines. On the other hand, the GTP-binding deficient K131A, which retains robust anti-HIV-1 activity in K562, HOS, U87-MG [6,7], and HeLa cells (S6 and S7 Figs), inhibited infection to a much lesser extent in HT1080 cells (less than two-fold), indicating that requirements for GTP binding are also cell-type specific. The requirement for GTP binding is also virus-specific, since the K131A mutation abrogated antiviral activity against HIV-2 and SIVmac, while all other MX2 mutants that inhibited HIV-1$_{WT}$ infection also inhibited HIV-2 and SIVmac infection.

In most cases, MX2 mutants that inhibited HIV-1$_{WT}$ infection also enhanced HIV-1$_{G89V\ CA}$ infection in both HeLa and HT1080 cells (MX1$_{MX2-NTD}$, SSS$_{14,17,18}$AAA, YRGK$_{487-490}$AAAA) while those that did not inhibit the WT virus had no effect on the G89V CA mutant virus (ΔN25, RRR$_{11-13}$AAA, M574D, R455D) (Figs 3, S6, and S7). Similarly, only MX2 mutants that inhibited HIV-1$_{WT}$ infection (MX1$_{MX2-NTD}$ and SSS$_{14,17,18}$AAA) were able to reverse the reduction in HIV-1$_{N57S\ CA}$ infectivity in the presence of CsA in HT1080 cells. These data suggest that similar determinants are involved in the antiviral activity of MX2, as well as the CA-specific, CypA-dependent pro-viral effects of MX2. The MX2$_{T151A}$ mutant, however, was a notable exception to this trend in that it failed to rescue HIV-1$_{N57S\ CA}$ infection in CsA-treated HT1080 cells, and inhibited HIV-1$_{G89V\ CA}$ infection in both HeLa and HT1080 cells. Thus, while CA-CypA interactions are required for the full antiviral activity of WT MX2, MX2 which can bind but not hydrolyze GTP restricts HIV-1 infection even in the absence of CypA binding.

We next tested the effects of GTPase-deficient MX2 on HIV infection in our CypA knockout and mutant cell lines. As expected, the MX2$_{T151A}$ mutant inhibited both HIV-1$_{WT}$ and HIV-1$_{G89V\ CA}$ infection in the presence or absence of CsA in control, CypA$^{-/-}$, and all CypA mutant cell lines (S8 and S9 Figs). Furthermore, MX2$_{T151A}$ failed to enhance HIV-1$_{N57S\ CA}$ infection in CypA$^{-/-}$, huCypA$_{D66N}$ and CypA$_{D66N/R69H}$ cells. Sensitivity of HIV-2 to both WT and T151A mutant MX2 was similar in all cell lines, further demonstrating that CypA-interactions with HIV-2 CA do not affect MX2 sensitivity (S8 and S9 Figs).

In 2022, Betancor, *et al.* [21] identified several serine and threonine residues of MX2 in addition to positions 14, 17, and 18 whose phosphorylation state affects the antiviral activity and substrate specificity of MX2. In particular, they found that MX2 T151 is phosphorylated, and demonstrated that a phosphomimetic MX2$_{T151D}$ exhibited increased antiviral activity

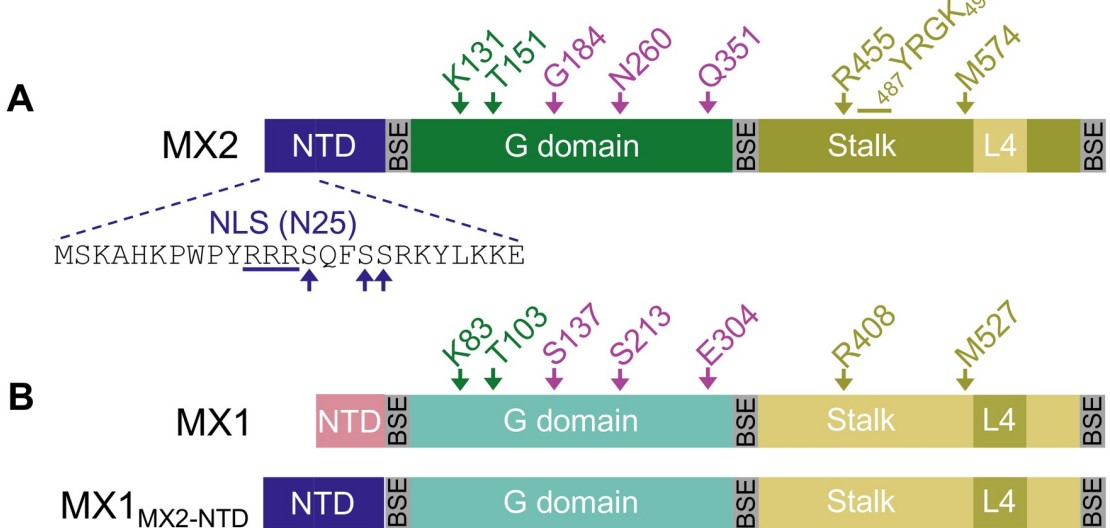

**C**

| Residues/mutation | Function/effect of mutation | References |
|---|---|---|
| **N91** | N-terminal domain<br>Can confer antiviral activity to heterologous proteins | (13, 17, 30) |
| **N25** | NPC targeting<br>CA binding | (5-7, 15-16) |
| **RRR$_{11-13}$** | Interaction with MLCP<br>CA recognition | (15, 20, 39, 40) |
| **SSS$_{14,17,18}$** | Phosphorylation suppresses function and reduces NPC localization<br> AAA$_{14,17,18}$ - Non-phosphorylatable<br> DDD$_{14,17,18}$ - Phosphomimetic | (20) |
| **T151** | GTPase activity<br>Phosphorylation affects antiviral activity and specificity<br> T151A – No GTP hydrolysis, non-phosphorylatable<br> T151D – Predicted to not bind GTP, phosphomimetic<br>Corresponds to MX1 T103 | (5, 21, 60, 61) |
| **K131A** | No GTP Binding<br>Corresponds to MX1 K83 | (5) |
| **M574D** | Monomeric<br>Corresponds to MX1 M527 | (18) |
| **R455D** | Limited oligomerization<br>Corresponds to MX1 R408 | (18) |
| **YRGK$_{487-490}$-AAAA** | Dimeric | (16, 18) |
| **G184/N260/Q351** | CA binding<br>Correspond to MX1 S137/S213/E304 | (40) |

**Fig 2. Summary of MX1 and MX2 mutants used in this study.** A) Diagram representing the various domains of MX2, with the N-terminal 25 amino acids shown in detail. BSE: bundle-signaling element. Residues of particular interest are indicated. B) Diagram representing the domains of MX1 (top), with residues corresponding to those highlighted in MX2 indicated, and diagram of chimeric fusion of MX1 containing the N-terminal 91 amino acids of MX2 (bottom). C)

Overview of mutations in MX2 known to affect CA-binding [5–8,15–17,19,20,39,40], phosphorylation 20,21], GTPase activity [5,21,60,61], and oligomerization [5,16,18]. Known consequences of mutations/functions of specific residues and relevant citations are shown.

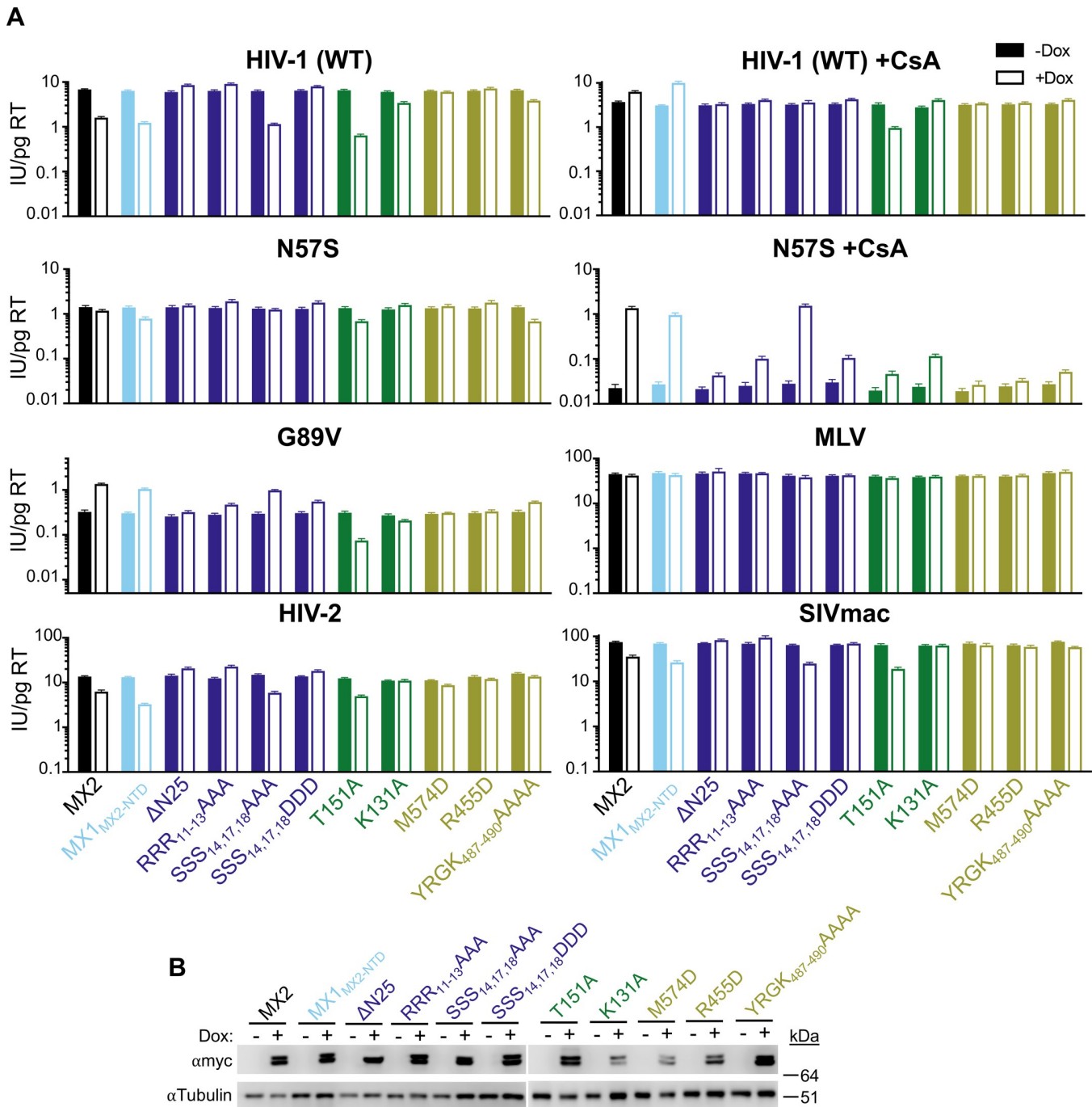

**Fig 3. Determinants for MX2 activity in the absence of CA-CypA interactions.** A) Infectivity of GFP reporter viruses in HT1080 cells expressing doxycycline-inducible C-terminally myc-tagged MX2, MX2 mutants, or MX1$_{MX2-NTD}$ in the presence (open bars) and absence (filled bars) of doxycycline (Dox). MX2 mutants are color-coded by domain/effect as in Fig 2A. Cells were infected in the presence of CsA where indicated. Titers are represented as mean + sem of infectious units (IU) per pg of reverse transcriptase (RT), n≥8 technical replicates combined from two-six independent experiments. Statistical analysis in S1 File. B) Western blot analysis of doxycycline-inducible MX2-myc and tubulin loading control.

against HIV-1$_{WT}$ and conferred antiviral activity against the MX2-resistant HIV-1$_{N74D\ CA}$, HIV-1$_{P90A\ CA}$, and HIV-1$_{T210K\ CA}$. To determine whether the ability of the MX2$_{T151A}$ mutant to restrict HIV-1 infection in the absence of CypA-CA binding is due to the lack of a phosphate group at this residue or due to the inability to hydrolyze GTP, we assessed the antiviral activity of the phosphomimetic T151D MX2 mutant in HT1080 and HeLa cells (Figs 4, S10, and S11). MX2$_{T151D}$ exhibited increased antiviral activity against HIV-1$_{WT}$, and to a smaller extent against HIV-2 and SIVmac in both cell types. While the previous report indicated that MLV infection in U87-MG cells is sensitive to MX2$_{T151D}$, we observed no restriction of MLV by this mutant. MX2$_{T151D}$ inhibited HIV-1$_{N57S\ CA}$ infection in both HeLa and HT1080 cells, and only partially rescued infectivity of this mutant in CsA-treated HT1080 cells (Figs 4, S10, and S11). Finally, similar to MX2$_{T151A}$, MX2$_{T151D}$ inhibited HIV-1$_{WT}$ infection in the presence of CsA as well as infection of HIV-1$_{G89V\ CA}$, demonstrating that phosphorylation at residue

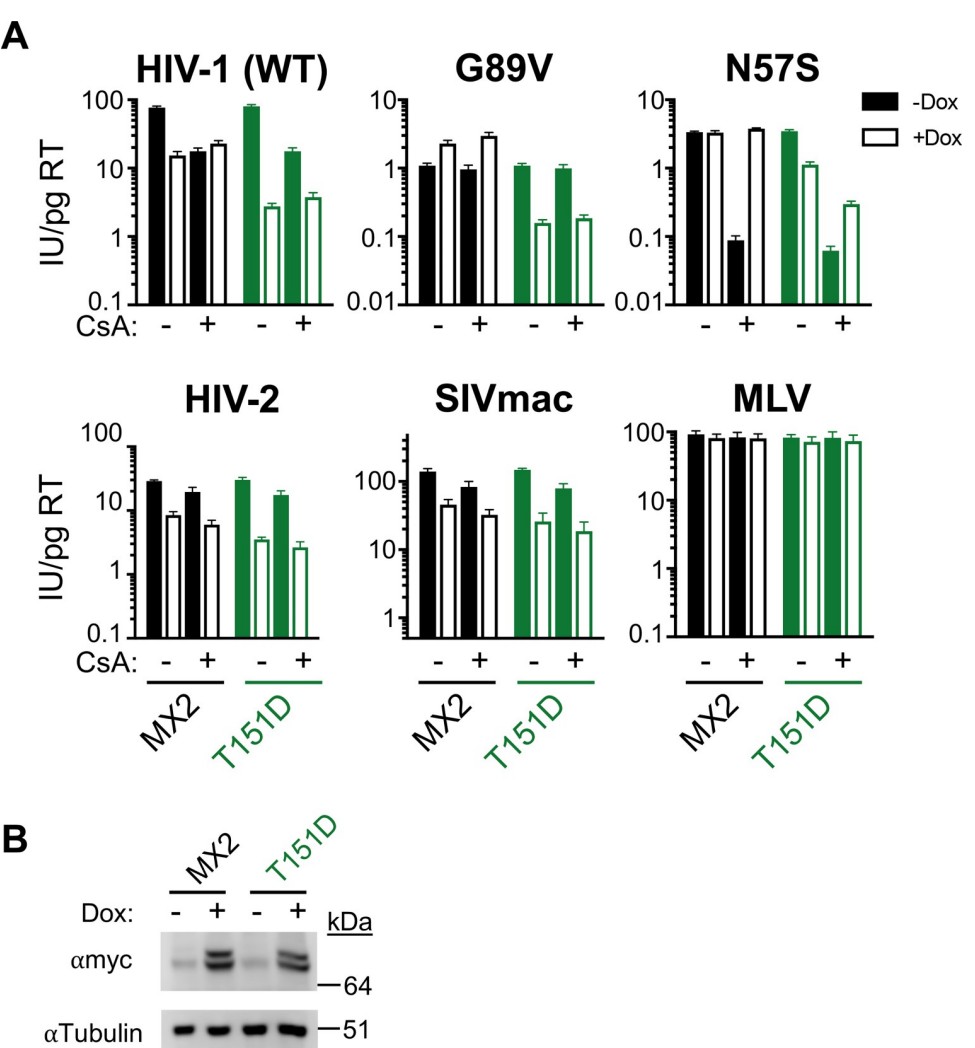

**Fig 4. Phosphorylation at residue T151 does not determine CypA-independent MX2 activity.** A) Infection HT1080 cells stably transduced with doxycycline-inducible myc-tagged MX2 or MX2$_{T151D}$ in the presence (open bars) or absence (filled bars) of doxycycline and presence or absence of CsA with the indicated GFP reporter viruses. Titers are represented as mean + sem of infectious units (IU) per pg of reverse transcriptase (RT), n≥9 technical replicates combined from three-five independent experiments. Statistical analysis in S1 File. B) Western blot analysis of doxycycline-inducible MX2-myc and tubulin loading control.

T151 does not determine CA-CypA-dependent antiviral activity of MX2 since $MX2_{T151A}$ and $MX2_{T151D}$ exhibited a similar phenotype. Collectively, these results indicate that the inability to hydrolyze GTP abrogates the requirement for HIV-1 CA-CypA interactions in the antiviral activity of MX2.

## Effect of GTPase domain-CA interactions on MX2 activity in the absence of CypA-CA binding

One possibility explaining the ability of T151A/D mutants of MX2 to inhibit HIV-1 infection in the absence of CA-CypA interactions is that the lack of GTP hydrolysis facilitates interactions between the viral CA and the CA-binding site of the GTPase domain [40]. To address this hypothesis, we constructed a number of MX2 mutants affecting GTPase activity (T151A), the N-terminal triple arginine motif ($_{11}AAA_{13}$) and/or the CA-binding residues in the GTPase domain [G184S, N260S, Q351E (SSE)] and assessed their antiviral activity (Figs 5, S12, and S13). Mutation of the triple arginine motif abrogated the antiviral activity of $MX2_{T151A}$ against HIV-1$_{WT}$, HIV-1$_{G89V\ CA}$, HIV-2, and SIVmac. On the other hand, mutation of the CA-binding residues of the GTPase domain did not affect the antiviral activity of $MX2_{T151A}$ in the presence or absence of CsA against these viruses; in fact, the SSE/T151A mutant appeared to have enhanced antiviral activity as compared to WT or T151A MX2. Therefore, these CA-GTPase domain interactions do not mediate the antiviral activity of GTPase-deficient MX2. Interestingly however, mutation of GTPase domain-CA interacting residues did affect the enhancement of HIV-1$_{N57S\ CA}$ infection in CsA-treated HT1080 cells, almost fully restoring the ability of $MX2_{T151A}$ to rescue HIV-1$_{N57S\ CA}$ infection, suggesting that the lack of enhancement by $MX2_{T151A}$ under these conditions is at least partially mediated by GTPase domain-CA interactions.

Next, we explored whether additional determinants affect the ability of the GTPase-deficient MX2 to restrict infection of primate lentiviruses in the absence of CA-CypA binding. We generated chimeric $MX1_{MX2-NTD}$ proteins with mutations corresponding to K131A (K83A) and T151A/D (T103A/D), as well as mutations in MX1 alone as a control, expressed them in HT1080 and HeLa cells, and assessed their antiviral activity (Figs 6, S14, and S15). Inhibition of HIV-2 and SIVmac infection by chimeric $MX1_{MX2-NTD}$ was unaffected by all three mutations in both cell lines, while GTP binding was required for restriction by MX2 (Figs 3, S6, and S7). The effect of these mutations on restriction of HIV-1$_{WT}$ was cell-type specific, as $MX1_{MX2-NTD}$ chimeras restricted HIV-1$_{WT}$ infection in the presence or absence of CsA in HeLa cells but did not restrict HIV-1$_{WT}$ in CsA-treated HT1080 cells. The effects on HIV-1$_{G89V\ CA}$ infection were also cell-type specific, while $MX1_{MX2-NTD}$ enhanced HIV-1$_{G89V\ CA}$ infection in both cell types, the K83A and T103A $MX1_{MX2-NTD}$ mutants modestly restricted HIV-1$_{G89V\ CA}$ infection. Furthermore, the T103D $MX1_{MX2-NTD}$ mutant had no effect on HIV-1$_{G89V\ CA}$ in HT1080 cells, while the opposite (no inhibition or enhancement by K83A $MX1_{MX2-NTD}$ and T103A $MX1_{MX2-NTD}$, restriction by T103D $MX1_{MX2-NTD}$) was observed in HeLa cells. Finally, HIV-1$_{N57S\ CA}$ infection was not inhibited by any $MX1_{MX2-NTD}$ chimeras in either cell type, and was rescued from the inhibitory effect of CsA treatment in HT1080 cells by all three GTPase $MX1_{MX2-NTD}$ mutants. The effect of GTPase mutations in $MX1_{MX2-NTD}$ chimeras on the HIV-1$_{N57S\ CA}$ mutant are in agreement with those shown in Fig 5 regarding the role of CA-interacting residues in the GTPase domain. However, the altered effects of GTP binding/hydrolysis mutant-$MX1_{MX2-NTD}$ chimeras on HIV-1$_{WT}$, HIV-1$_{G89V\ CA}$, HIV-2, and SIVmac infection as compared to the K131A, T151A, and T151D mutants of MX2 demonstrate that additional unknown determinants in the GTPase, stalk, or L4 domain affect interactions between MX2 and these viruses.

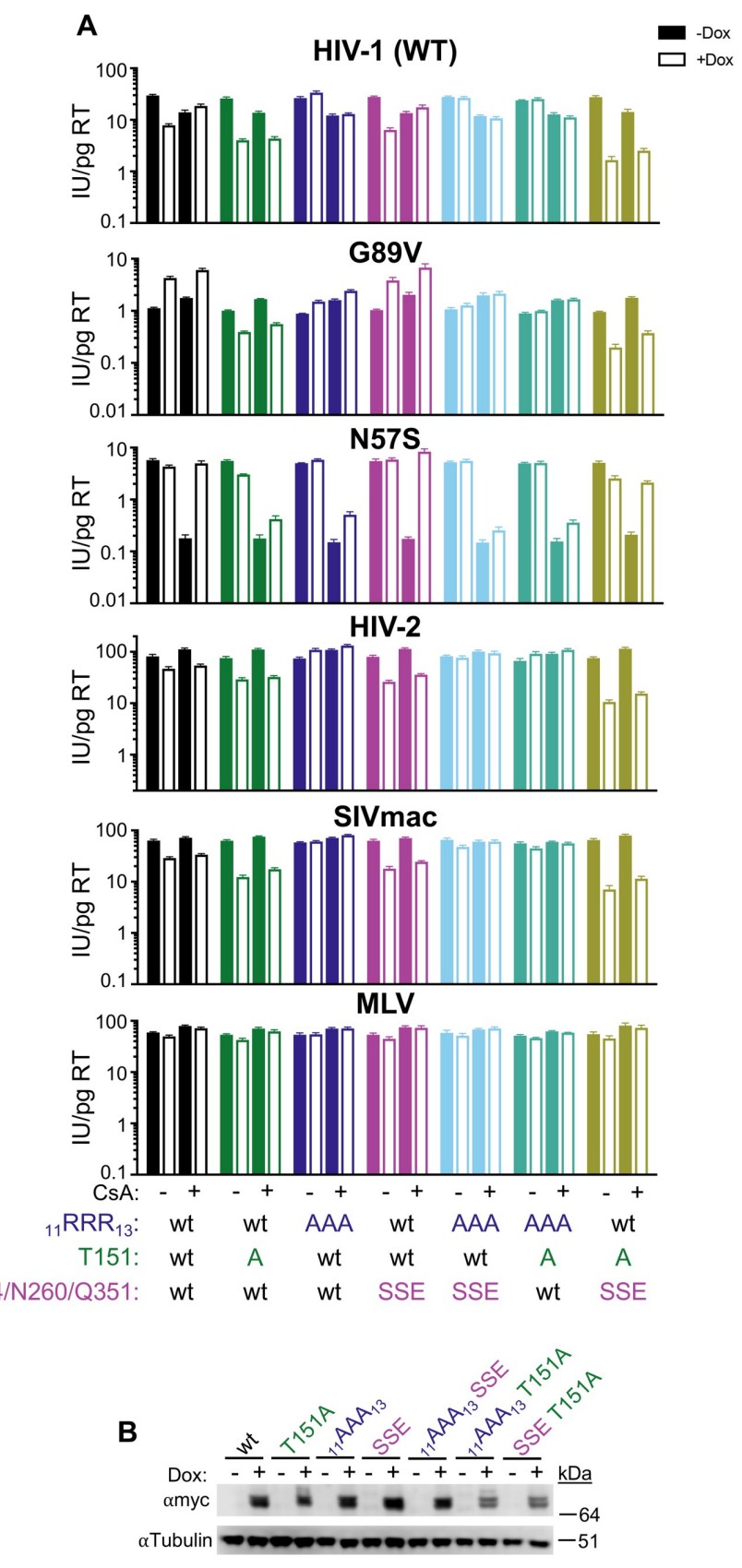

**Fig 5. Restriction by GTPase-deficient MX2 in the absence of CA-CypA binding is not mediated by known CA-GTPase domain interactions.** A) Infection of HT1080 cells stably transduced with doxycycline-inducible myc-tagged MX2 with or without mutations in the N-terminal triple-arginine motif ($_{11}$AAA$_{13}$), T151A, or CA-binding residues in the GTPase domain (SSE) in the presence (open bars) or absence (filled bars) of doxycycline and presence or absence of CsA with the indicated GFP reporter viruses. Titers are represented as mean + sem of infectious units (IU) per pg of reverse transcriptase (RT), n≥9 technical replicates combined from three-five independent experiments. Statistical analysis in S1 File. B) Western blot analysis of doxycycline-inducible MX2-myc and tubulin loading control.

## Effects of Nup and NTR depletion on the antiviral activity of GTPase-deficient MX2

Our previous work demonstrated Nups mediate the antiviral activity of MX2 in a CA and CypA-dependent manner [30]. Since MX2$_{T151A}$ retains antiviral activity in the absence of CA-CypA interactions, we next sought to determine whether this mutation also affected Nup/NTR requirements for antiviral function. Using our previously validated siRNA library targeting Nups and NTRs (Fig 7A and [30]), HT1080 cells were transfected with siRNA before being split into replicated wells for MX2$_{WT}$ or MX2$_{T151A}$ induction (via doxycycline addition) followed by infection with HIV-1$_{WT}$ (S16 Fig). Knockdown of either RANBP2 or NUP153 abrogated the antiviral activity of both MX2$_{WT}$ and MX2$_{T151A}$ against HIV-1$_{WT}$ (Fig 7B and 7C). Furthermore, depletion of the Nup62 subcomplex, transmembrane, and nuclear basket Nups also had similar effects (eg. NUP62 depletion reduced MX2$_{WT}$ and MX2$_{T151A}$ activity, while NUP54, NUPL1, POM121, NDC1, NUP50, and TPR depletion either increased antiviral activity or had no effect), indicating that the T151A MX2 mutation does not completely alter MX2-Nup interactions. Moreover, the effects of NTR depletion were similar, suggesting that abrogation of GTPase activity does not affect MX2-transportin interactions. However, a number of differences were evident in the Nup requirements for MX2$_{WT}$ and MX2$_{T151A}$ activity (eg. multiple components of the Nup107 and Nup93 subcomplexes reduced the antiviral activity of MX2$_{WT}$, but not MX2$_{T151A}$). Most notably, NUP88 and NUP214 depletion abolished the antiviral activity of MX2$_{WT}$, but not MX2$_{T151A}$.

We next investigated the effects of Nup depletion on the MX2 sensitivity of HIV-1$_{G89V\ CA}$. As we previously reported [30], the increase in HIV-1$_{G89V\ CA}$ infectivity exerted by MX2 is amplified by some Nup depletions (eg. NUP62, NUP88, NUP214, NUP153), and diminished by others (eg. NUP155, NUP54, RANBP2) (Fig 7D). As expected, a number of Nup depletions also altered the activity of MX2$_{T151A}$ against HIV-1$_{G89V\ CA}$ (Fig 7E). We observed no clear correlation between whether a knockdown affected enhancement of HIV-1$_{G89V\ CA}$ infection by MX2 and whether it caused loss of MX2$_{T151A}$ activity against HIV-1$_{G89V\ CA}$. Depletion of transmembrane, cytoplasmic, and nuclear basket Nups, as well as NTRs had similar effects on the antiviral activity of MX2$_{T151A}$ against both HIV-1$_{WT}$ and HIV-1$_{G89V\ CA}$ (Fig 7C and 7E). Conversely, depletion of several Nups (SEC13, NUP107, NUP133, NUP205, NUP155, NUP52, NUPL1) affected restriction of HIV-1$_{G89V\ CA}$ but not HIV-1$_{WT}$, while only NUP98 knockdown reduced antiviral activity against HIV-1$_{WT}$ but not HIV-1$_{G89V\ CA}$. Thus, these data indicate that the antiviral activity MX2$_{T151A}$ has distinct requirements for Nups and NTRs, some of which are also affected by CA-CypA interactions.

## Discussion

While the NTD of MX2 is of principal importance for viral substrate recognition and interaction with cellular proteins [6,9–13,15,17,39], there is a growing appreciation for the importance of the GTPase domain in MX2 function [21,40,41]. Here, we demonstrate that the GTPase activity of MX2 affects substrate specificity and nucleoporin requirements for anti-HIV-1 activity. While WT MX2 does not restrict HIV-1 in the absence of CypA binding to

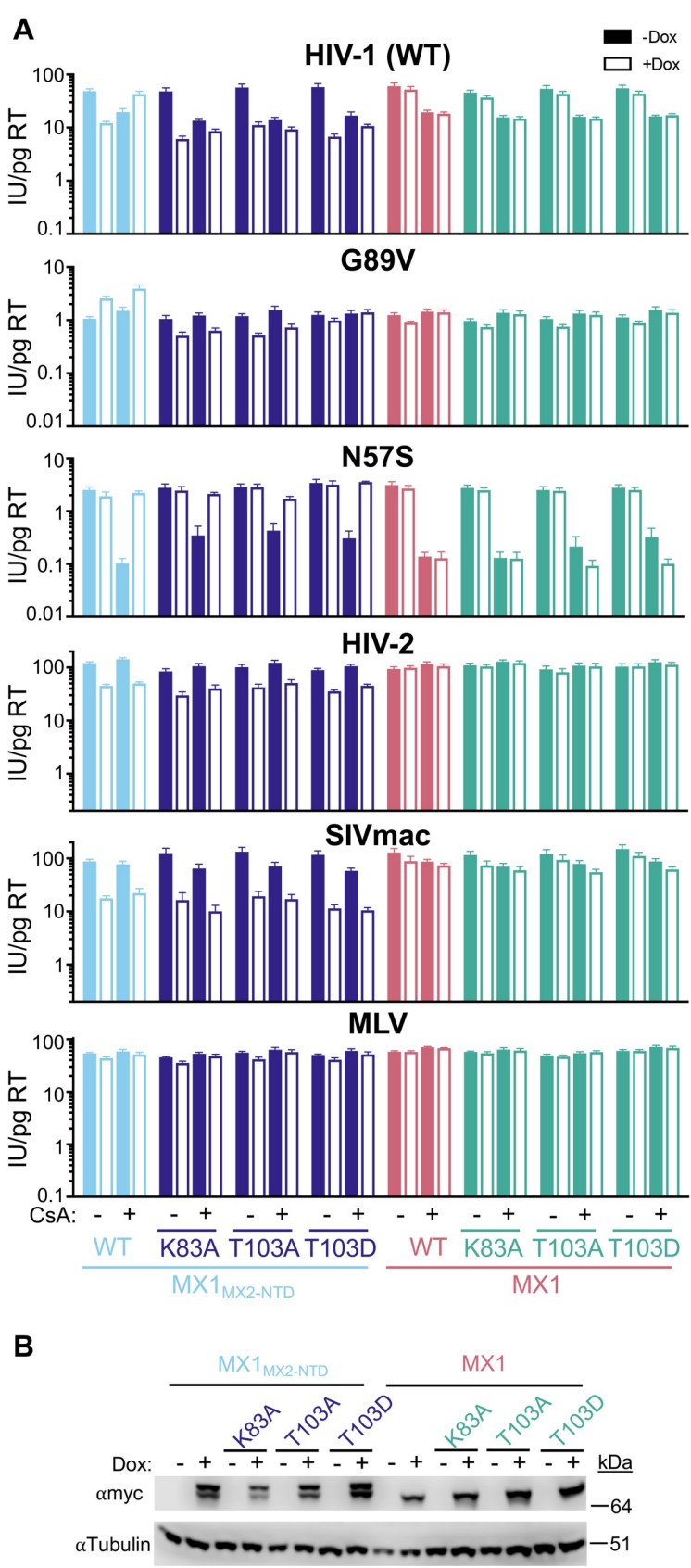

**Fig 6. Antiviral activity of GTPase-deficient chimeric MX1$_{MX2-NTD}$ proteins.** A) Infection of HT1080 cells stably transduced with doxycycline-inducible myc-tagged MX1 or MX1$_{MX2-NTD}$ with the indicated mutations in the presence (open bars) or absence (filled bars) of doxycycline and presence or absence of CsA with the indicated GFP reporter viruses. Titers are represented as mean + sem of infectious units (IU) per pg of reverse transcriptase (RT), n≥9 technical replicates combined from three-seven independent experiments. Statistical analysis in S1 File. B) Western blot analysis of doxycycline-inducible MX1$_{MX2-NTD}$ or MX1-myc and tubulin loading control.

CA, the T151A MX2 mutant, which is able to bind, but not hydrolyze GTP [5], retains antiviral activity in the absence of CA-CypA binding (Figs 3, S5, S6, and S7). By generating cells containing endogenous mutations in CypA that affect CA recognition, we demonstrate that the effects of CypA (and CsA treatment) on HIV-1 and HIV-1 CA mutant infection and MX2 sensitivity are the direct result of interactions with CA, and do not appear to involve interactions between CypA and other cellular proteins (Figs 1, S2, S4, and S5).

While the GTPase domain of MX2 has been found to directly bind the HIV-1 CA [40], we found that these interactions were not involved in the CypA-independent activity of GTPase-deficient MX2 (Figs 5, S12, and S13). However, the antiviral activity of GTPase-deficient chimeric MX1$_{MX2-NTD}$ fusions did not recapitulate the phenotype of the T151A mutant (Figs 6, S14, and S15). Furthermore, MX1$_{MX2-NTD}$ increased HIV-1 infectivity in CsA-treated HT1080s to a greater degree than MX2 (Figs 3, S7, and S15) therefore there are additional determinants unique to MX2 that affect phosphorylation, oligomerization/conformation, CA recognition, or interaction with other cellular proteins which alter antiviral activity and substrate specificity. Importantly, we found that nucleoporin requirements of the T151A MX2 mutant for antiviral activity were altered as compared to WT MX2 in both CypA-dependent and -independent ways, suggesting that this mutation affects both antiviral substrate-specificity, and interaction with nuclear pore complexes. Additionally, since MX2 also affects nuclear import of some non-viral cargos, it will be of interest in future investigations to determine whether the determinants identified here which affect viral specificity also alter inhibition of specific nuclear import pathways.

Interestingly, restriction of HIV-2, SIVmac, and SIVagmTAN by MX2 does not involve CA-CypA interactions ([30], Figs 1, S4, and S5), and GTP hydrolysis is dispensable for inhibition of HIV-2 and SIVmac by both MX2 and MX1$_{MX2-NTD}$ fusions (Figs 3, 6, S6, S7, S14, and S15). Therefore, it is likely that recognition of HIV-1 and other primate lentiviruses by MX2 involve distinct interactions with the capsid surface, consequently requiring distinct determinants. Indeed, given the ability of MX2 to inhibit replication of highly divergent viruses, including herpesviruses and flaviviruses (reviewed in [22]), it appears that we are just beginning to understand the mechanisms regulating MX2 specify and activity.

We also identified here a number of MX2 determinants that are cell-type specific, highlighting the importance of other cellular players in MX2 activity. While our previous investigation found that the K131A mutant of MX2 which is unable to bind GTP [5], retains antiviral activity [6,7], here we found that this activity is cell-type dependent since this mutant is less potent in HT1080 cells (Fig 3). We further find that the YRGK$_{487-490}$AAAA mutant of MX2, has minimal antiviral activity in either HT1080 or HeLa cells, demonstrating that the requirements for higher-order oligomerization of MX2 are also cell-type dependent. While the MX2 stalk is the major driver of oligomerization, the GTPase domain also exerts effects on MX2 conformation [41]. Biochemical studies have found that MX2 spontaneously forms helical assemblies that are depolymerized into oligomers upon GTP binding; importantly, the T151A MX2 mutant also self-assembles, however these assemblies are loosened, but not dissociated upon GTP binding [41]. Thus, conformational changes in MX2 may explain the expanded substrate-specificity of the T151A mutant. Importantly, recent work has shown that the MX2 residue T151 is

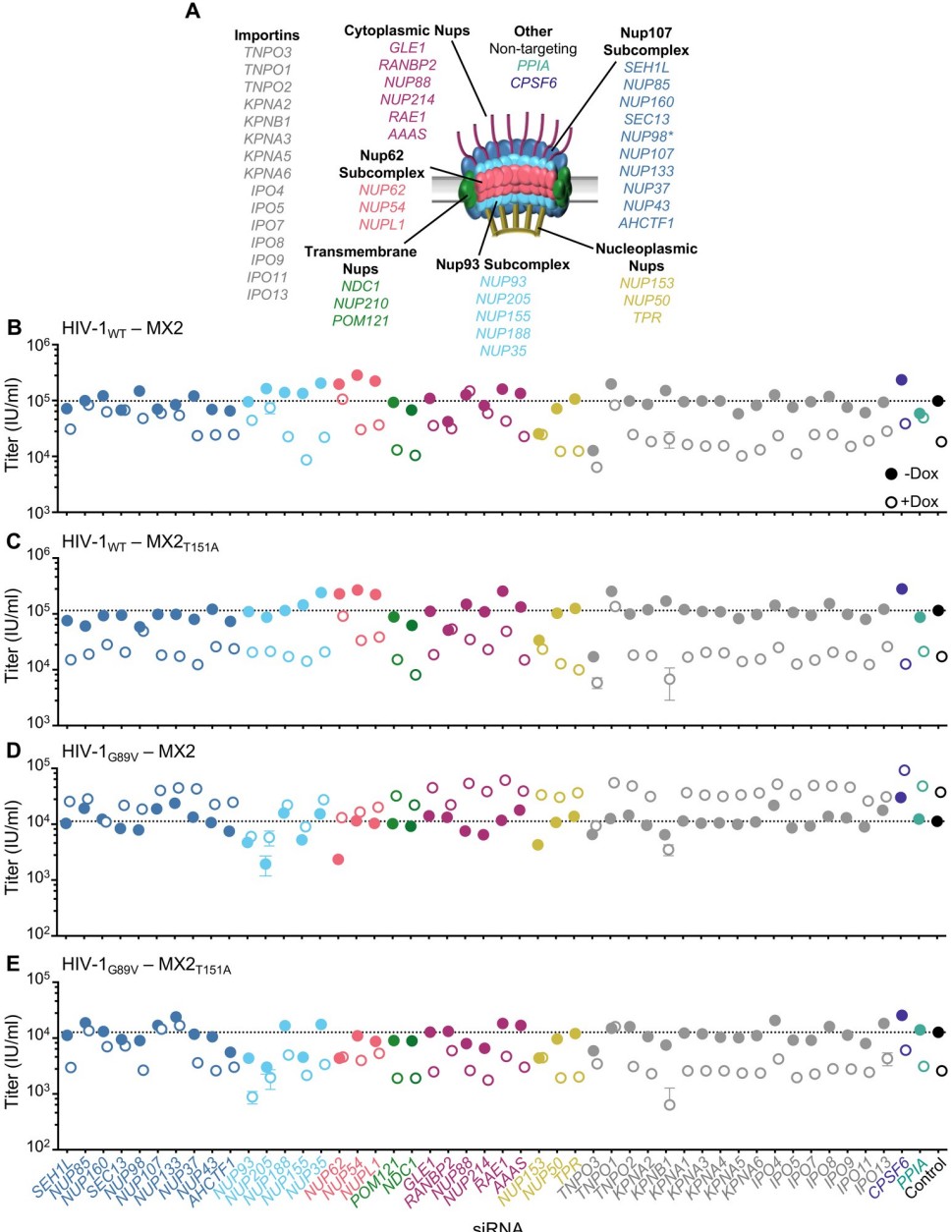

**Fig 7. GTPase-deficient MX2 has distinct nucleoporin requirements for antiviral activity.** A) Schematic representation of the nuclear pore complex and genes included in siRNA library color coded by subcomplex. (*-NUP98 is listed as a member of Nup107 subcomplex, however. Nup98 and Nup96 are produced following autoproteolytic cleavage of a polyprotein precursor [62,63], this siRNA will target both Nups). Importins/nuclear transport receptors (NTRs) included in the siRNA library listed in grey. Also included, siRNA targeting CypA, CPSF6, and a non-targeting control siRNA. B-D) Infectivity of HIV-1 GFP reporter viruses in HT1080 cells stably transduced with doxycycline-inducible MX2 or MX2$_{T151A}$ in the presence (open circles) or absence (filled circles) of doxycycline 64-72h after transfection with siRNA (color coded by subcomplex as in A). Titers are mean ± sem, n = 3 technical replicates, representative of four independent experiments. Statistical analysis in S1 File.

phosphorylated, and that the phosphomimetic T151D mutant has increased antiviral activity [21]. We recapitulated this finding and additionally found that this mutant is also able to restrict HIV-1 infection in the absence of CA-CypA interactions (Figs 4, S10, and S11),

indicating that the lack of phosphorylation of the T151A mutant does not explain its expanded substrate specificity. While the T151D MX2 mutant is unlikely to bind GTP, as noted by Betancor, G. *et al*, the negative charge of the phosphomimetic T151D mutant may resemble the GTP-bound T151A mutant, and both mutants have distinct subcellular localization as compared to WT MX2 [21], they may both therefore exist in a "loose helical assembly". While high GTP concentrations within cells would suggest that MX2 exists in its GTP-bound non-assembled state [41], it is also possible that changes in the phosphorylation state of T151 regulates MX2 conformation (and GTP binding), thereby affecting interactions with both viral substrates and nuclear pore complexes.

A particularly intriguing cell-type dependent effect is the dramatic loss in infectivity of HIV-1$_{N57}$ CA mutants in the absence of CypA binding in HT1080, but not HeLa cells [30], which is then reversed by WT, but not GTPase-deficient MX2 (Figs 1, 3, S4, S5, S6, and S7). One possible explanation for the phenotype in HT1080 cells is that MX2 shields HIV-1$_{N57}$ CA mutants (and to a lesser extent HIV-1$_{WT}$ and HIV-1$_{G89V\ CA}$) from TRIM5alpha; restriction in a manner that depends on GTP hydrolysis. This would further suggest that conformational dynamics of HIV-1 CA (as proposed by [31]) as well as MX2 affect viral sensitivity to TRIM5α. The distinct phenotype in HeLa cells could be explained by lower expression levels of HIV-1 insufficient for restriction, or by differences in additional cellular factors affecting HIV-1 CA or TRIM5α (eg. TRIM34 [42,43]).

Collectively, the current data support a model in which GTP binding and hydrolysis effect conformational changes in MX2 that govern interactions with the viral CA. Given evidence suggesting that CypA-dependent changes in HIV-1 CA produce an MX2 sensitive conformation or state [31], it appears that restriction by MX2$_{T151A}$ is independent of CA conformation since sensitivity of HIV-1 to MX2$_{T151A}$ is CypA-independent (although we cannot definitively exclude the possibility that CypA isomerizes an unknown cellular protein that interacts with CA). Our model would further indicate that spatiotemporal effects of interactions between CA, CypA, and MX2 impact interactions with other cellular co-factors (such as Nups) and/or restriction factors (eg. TRIM5α), which ultimately determines the outcome of infection.

In summary, this report highlights the importance of GTPase activity in modulation of the substrate specificity of MX2 and further indicate that highly complex interactions between MX2 and its cellular environment determine antiviral activity. While we have focused specifically on human MX2, the regulation of the antiviral activity of other MX proteins is similarly complex. Given the importance of MX proteins in the innate immune responses of a highly divergent range of vertebrate species [3], the evolution of *Mx* genes has likely been the result of complex selective pressures balancing substrate breadth and specificity, as well as interaction with cellular partners.

## Materials and methods

### Plasmid construction

**MX2.** MX1 and MX2 mutants with or without a C-terminal myc-tag were cloned into the doxycycline-inducible vector pLKOΔ [13,30] using *Sfi*I. MX2ΔN25, MX2$_{T151A}$, and MX2$_{K131A}$ were PCR amplified from pCSIB expression vectors [6], MX2$_{G184S/N260S/Q351S}$ (SSE) and MX2-SSE$_{T151A}$ were synthesized as gBlock fragments [Integrated DNA Technologies (IDT)], other mutants were generated via PCR mutagenesis using the oligos indicated in S1 Table.

**TRIM5-fusions.** An LHCX MLV vector (Clontech) was engineered to express the N-terminal domains (RIGN, B-Box and coiled coil) from owl monkey (omk) TRIMCyp followed by an HA tag and cloning sites (NotI/SalI) that allowed introduction of various proteins as described in [44,45]. The plasmids generated expressed HA-TRIM5 N-terminus fused to the

Cyclophilin A domain from owl monkey TRIMCyp (omkTRIMCyp), the human Cyclophilin A protein (huCypA), or the human Cyclophilin A protein with the D66N, R69H, or D66N/R69H (pigtail macaque TRIMCyp) mutations generated by overlap PCR with the primers indicated in S1 Table.

**AAV donor.** To construct the pAAV-*PPIA* donor plasmids for the generation of CypA point mutant cell lines (S2 Fig), a 1596bp region of the *PPIA* locus (bases 7161 to 8757 NCBI Reference Sequence: NG_029697.1) was synthesized as a gBlock fragment (IDT) and inserted into the pAAV packaging plasmid (Cell Bio Labs) using *Nhe*I and *Xho*I. The PAM site for crRNA 2 was mutated from TGG to TGA and for crRNA 3 form CCA to CTA. A silent *Hpa*I site was introduced into intron 3 as shown in S2C Fig. The CypA$_{R69H}$ donor template was synthesized, then CypA$_{D66N}$ and CypA$_{D66N/R69H}$ donors were generated via overlap PCR with the primers indicated in S1 Table.

**HIV-1 CA mutants.** HIV-1 GFP reporter viruses containing the N57A or N57D capsid mutation (HIV-1$_{N57A}$ and HIV-1$_{N57D}$) was generated by overlap PCR (primers listed in S1 Table) and insertion into pNHGCapNM as described previously [32] using *Not*I and *Mlu*I sites.

**Cell lines.** HEK 293T, HeLa, and HT1080 cell lines were maintained in Dulbecco's Modified Eagles Medium [(DMEM) Gibco] with 10% fetal calf serum (Gibco), gentamicin (Gibco), and Plasmocin prophlyactic (InvivoGen). Plasmocin was not present in cultures during infection, transduction, or transfection. Cells were purchased from ATCC and were assumed to be authenticated by their supplier and were not further characterized. Cells were monitored quarterly for retroviral contamination by SYBR-Green based PCR RT assay [46,47] and tested for mycoplasma contamination by MycoStrip detection kit (InvivoGen). Derivatives of HeLa and HT1080 cells containing doxycycline-inducible Mx2, or fusion proteins were generated by transduction with LKO-derived lentiviral vectors [13] followed by selection in 1μg ml$^{-1}$ puromycin (Sigma-Aldrich). HeLa cells stably expressing TRIM fusions were generated by transduction with LHCX-derived vectors followed by selection in 100μg ml$^{-1}$ hygromycin B (Gibco). Vector stocks for transduction were generated by co-transfection of 293T cells with a VSV-G expression plasmid, an HIV-1$_{NL4-3}$ Gag-Pol expression plasmid, and LKO-derived vector, or an MLV Gag-Pol expression plasmid and an LHCX-derived vector using polyethyleneimine (PolySciences). Expression was induced in pLKO transduced cell lines through an overnight treatment with 500ng/ml doxycycline hyclate (Sigma-Aldrich) prior to challenge with retroviruses or retroviral vectors.

**Generation of CypA knock-out and mutant cell lines.** AAVs containing donor sequence for homology directed repair were generated by co-transfection of 293T cells pAAV-*PPIA*, pAAV-RC (DJ), and pHelper (CellBio Labs) at a 1:1:1 ratio using polyethyleneimine (PolySciences). Supernatant was collected 72hrs post-transfection. 1x10$^5$ HT1080 cells in a 24-well plate were infected with 200μL of AAV-containing supernatant 4–6 hours prior to reverse transfection with Cas9 ribonucleoprotein complexes (RNPs).

Custom Alt-R CRISPR Cas9 guide RNAs targeting the *PPIA* locus were designed using the IDT website. RNPs containing crRNA 1, 2, or 3:tracerRNA duplexes were generated according to manufacturer instructions: 1μL of 100 μM crRNA, 1μ of 100μM tracrRNA-ATTO 550 (IDT) in 98μL of Nuclease-Free Duplex Buffer (IDT) were heated at 98˚C for 5min, then cooled to room temperature. 15μL of annealed guide RNA oligos were mixed with 15μL of 1μM recombinant Cas9 (IDT) in 220μL of Opti-MEM media (Gibco) and incubated for 5min at room temperature, complexes were used immediately or stored at -80˚C. HT1080 cells were reverse transfected in 96-well plates with RNPs using RNAiMax (Thermo Fisher) according to manufacturer instructions, transfection complexes containing 15μL of each RNP was mixed with 1.2μL of RNAiMax and 18.8μL of OptiMEM for 20mins at room temperature, followed

by addition of $4\times10^4$ cells/well. For generation of point mutant cell lines, 1μM Alt-R HDR Enhancer V2 (IDT) was included. HT1080 cells were transfected with RNPs containing guides 1 and 3 to generate CypA$^{-/-}$ cells and guides 2 and 3 to generate CypA-mutant cells (S2 Fig). Control cells were transfected with RNPs containing the IDT negative control crRNA. 16hours post-transfection, cells were transferred to a 24-well plate, and AAV infection/transduction was repeated after an additional 24-48hours. Single-cell clones were then derived by limiting dilution.

CypA$^{-/-}$ clones were screened via western blotting, sequences of knock-out alleles were determined following PCR amplification from genomic DNA using (In1 F2 and In4 R primers), followed by cloning into pCR-Blunt II TOPO [Zero Blunt TOPO kit (Thermo Fisher)], and sequencing ≥3 clones from each cell line. CypA point mutant clones were screened via PCR amplification from genomic DNA using (In1 F and In4 R2 primers) followed by *Hpa*I digestion. Candidates were confirmed by PCR and sequencing from both genomic DNA and cDNA [using SuperScript III Reverse Transcriptase (Thermo Fisher)]. Genomic DNA and mRNA were extracted using NucleoSpin Tissue and NucleoSpin RNA kits (Macherey-Nagel), respectively.

All experiments included all CypA mutant and knock-out clones shown in S2D Fig, and two-three control clones. All figures show representative CypA$^{-/-}$ clone #44, CypA$_{D66N}$ #33, CypA$_{R69H}$ #50, CypA$_{D66N/R69H}$ #74, and Control clone #1.

**Viruses.**   All viruses were generated by transfection of 293T cells using polyethyleneimine (PolySciences). GFP reporter proviral plasmids HIV-1$_{NL4-3}$ΔEnv-GFP (HIV-1, HIV-1$_{G89V}$, HIV-1$_{A92E}$, and G94D CA mutants [48]), NHGCapNM [with NL4-3 CA (wild-type, HIV-1$_{N57S}$, HIV-1$_{N57A}$, HIV-1$_{N57D}$, HIV-1$_{N74D}$, G208R, and HIV-1$_{T210K}$ CA mutants (this investigation and [30,32])], HIV-2$_{ROD}$ΔEnv-GFP, SIV$_{MAC}$ΔEnv-GFP, SIV$_{AGM}$TANΔEnv-GFP [49] 10μg of proviral plasmid was co-transfected with 1μg of VSV-G expression plasmid. For MLV, EIAV, and FIV, three plasmid vector systems [50–53] were also used to generate GFP reporter viruses, whereby 5μg of Gag-Pol, 5μg of packageable genome, and 1μg of VSV-G expression plasmids were co-transfected. Levels of reverse transcriptase in viral stocks were quantified using a one-step SYBR-Green based PCR RT assay as previously described [46,47].

**Infection assays.**   Infectivity was measured in HeLa or HT1080 cells seeded in 96-well plates at $5 \times 10^3$ cells per well and inoculated with serial-dilutions of VSV-G pseudotyped GFP reporter viruses in the presence of 4μg ml$^{-1}$ polybrene (Sigma-Aldrich). Where indicated, cyclosporine A (Sigma-Aldrich) was added to the cultures at the time of infection at 5μM. Infected cells (%GFP positive of viable cells) were enumerated by FACS analysis using an Attune NxT coupled to an Autosampler (Invitrogen). Prior to experimentation, viruses were titered on control cells to determine the inoculum required for infectivity within the linear range. Infectivity is reported as either infectious units (IU)/ml, or normalized to levels of reverse transcriptase (IU/pg RT) to control for variability in viral preps.

**CA-binding assay with HIV-1 CA tubes.**   CA(WT), CA(G89V), and CA(N57A) were expressed from pET3a in BL21-DE3 cells and purified as previously described [54–56]. CA nanotubes were assembled in a high-ionic strength buffer (15 mM Tris-HCl, pH 8.0; 2 M NaCl) as described [57,58]. Binding assays with HIV-1 CA nanotubes were performed as described [30]. In brief, control and CypA mutant HT1080 cells were lysed by adding a passive lysis buffer (Promega) supplemented with protease inhibitor cocktail (Roche). NaCl concentration in lysates was adjusted to 2 M and lysates were centrifuged at 13,000 x g for 2 min at 4°C. Then the supernatant of the cell lysates was added to the preformed CA nanotubes and incubated at room temperature for 30 min. Following centrifugation at 13,000 x g for 2 min at 4°C, the supernatant was saved, and the pellet was washed three times with the high-ionic strength buffer. LDS Reducing Sample buffer (Thermo Scientific Chemicals) was added to

both pulled-down and unbound fractions, and they were subjected to SDS-PAGE. The proteins of interest were detected by immunoblotting using respective antibodies.

**RNA interference.** RNA interference using a custom siRNA library (Table 1) targeting Nups and NTRs was performed as previously described [30]. HT1080 cells stably transduced with doxycycline-inducible MX2 or $MX2_{T151A}$ were reverse transfected with 25pmol of siRNA (Table 2; SMARTpool, Dharmacon) using Lipofectamine RNAiMax (Invitrogen) at a concentration of $5 \times 10^4$ cells/ml in 12-well plates. Non-transfected cells, transfection reagent alone, and non-targeting siRNA were used as controls, no significant difference in viral infectivity or Mx2-restriction were observed for each of these controls, as such, for each experiment, only the non-targeting siRNA control is shown. 24 hours after transfection, cells were trypsinized, diluted 1:2.5 and re-plated in 96-well plates and treated with doxycycline, followed by infection with GFP reporter viruses 36-48h later (See S10B Fig).

**Table 1. ON-TARGET SMARTpool siRNA utilized in this investigation.**

| Gene Symbol | Gene ID |
| --- | --- |
| AAAS | 8086 |
| AHCTF1 | 25909 |
| CPSF6 | 11052 |
| GLE1 | 2733 |
| IPO11 | 51194 |
| IPO13 | 9670 |
| IPO4 | 79711 |
| IPO5 | 3843 |
| IPO7 | 10527 |
| IPO8 | 10526 |
| IPO9 | 55705 |
| KPNA1 | 3836 |
| KPNA2 | 3838 |
| KPNA3 | 3839 |
| KPNA4 | 3840 |
| KPNA5 | 3841 |
| KPNA6 | 23633 |
| KPNB1 | 3837 |
| NDC1 | 55706 |
| NUP107 | 57122 |
| NUP133 | 55746 |
| NUP153 | 9972 |
| NUP155 | 9631 |
| NUP160 | 23279 |
| NUP188 | 23511 |
| NUP205 | 23165 |
| NUP210 | 23225 |
| NUP214 | 8021 |
| NUP35 | 129401 |
| NUP37 | 79023 |
| NUP43 | 348995 |
| NUP50 | 10762 |
| NUP54 | 53371 |

(*Continued*)

**Table 1.** (Continued)

| Gene Symbol | Gene ID |
|---|---|
| NUP62 | 23636 |
| NUP85 | 79902 |
| NUP88 | 4927 |
| NUP93 | 9688 |
| NUP98 | 4928 |
| NUPL1 | 9818 |
| POM121 | 9883 |
| PPIA | 5478 |
| RAE1 | 8480 |
| RANBP2 | 5903 |
| SEC13 | 6396 |
| SEH1L | 81929 |
| TNPO1 | 3842 |
| TNPO2 | 30000 |
| TNPO3 | 23534 |
| TPR | 7175 |

**Table 2. Antibodies utilized in this investigation.**

| Reactivity | Species | Company | Catalog Number |
|---|---|---|---|
| GAPDH | mouse | Santa Cruz | sc-47724 |
| HIV-1 p55+p24+p17 | rabbit | Abcam | ab63917 |
| MX2 | rabbit | Novus Biologicals | NBP1-8108 |
| Myc tag | mouse | Millipore | 05–724 |
| CypA | rabbit | Proteintech | 10720-1-AP |
| HA tag | Mouse | BioLegend | 901514 |
| Tubulin | mouse | Sigma-Aldrich | T6074 |

**Western blotting.** Cell suspensions were lysed in NuPage LDS (Invitrogen) or SDS sample buffer, followed by sonication, and separated by electrophoresis on NuPage 4–12% Bis-Tris gels (Invitrogen) and blotted onto polyvinylidene fluoride (PDVF, BioRad Laboratories). Membranes were incubated with the antibodies listed in Table 2, followed by incubation with goat anti-rabbit-HRP or goat anti-mouse-HRP secondary antibodies (Jackson ImmunoResearch). SeeBlue Plus2 and MagicMark XP Pre-stained Protein Standards (Thermo Fisher) were used. Blots were developed with SuperSignal West Femto Maximum Sensitivity Substrate (Thermo Fisher) and imaged on a C-Digit scanner (LI-COR Biosciences).

**Statistical analyses.** Statistical significance was determined using GraphPad software (two-way ANOVA). Statistical analyses for all figures are included in S1 File.

## Supporting information

**S1 Table. PCR primers.** Oligonucleotides used for cloning and genomic DNA amplification. (PDF)

**S1 File. Statistical Analysis.** (XLSX)

**S1 Fig. Recognition of retroviral capsids by TRIMCyp fusions.** A) Infectivity of GFP reporter viruses on HeLa cells stably expressing control empty vector, owl monkey TRIMCyp (owmTRIMCyp), or chimeras of the TRIM5 N-terminal domain with human cyclophilin A (huCypA), or human CypA$_{D66N}$, CypA$_{R69H}$, CypA$_{D66N/R69H}$ mutants. Titers are represented as mean + sem of infectious units (IU) per mL, n = 3 technical replicates representative of five independent experiments. Statistical analysis in S1 File. B) Expression of the TRIM fusion proteins tagged with HA and tubulin loading control in stable cell lines used in (A).
(PDF)

**S2 Fig. Generation of CypA knockout and mutant cell lines.** A) Schematic of the *PPIA* (CypA) locus on Chromosome 7 with crRNA guide targeting sites intron 1 and exon 4, and PCR primers for clone screening and verification indicated. Diagram of AAV donor for homology-directed repair shown below, as well as amino acid residues 66–69 in wild-type and mutant CypA cells. B) Sequences of the wild-type locus at targeted sites, with crRNA and PAM sites indicated. Dots indicate sequences flanking those shown in detail. Allele sequences of CypA$^{-/-}$ clones with mismatches to wild-type sequence highlighted and deleted sequence indicated by dashes, consequence of mutations in each knock-out allele shown below. C) Sequences of wild-type and point-mutant alleles with nucleotide changes in codons and *Hpa*I site in intron 3 indicated. D) Top: Infectivity of HIV-1 GFP reporter virus infection in control, CypA$^{-/-}$ (left), and CypA-mutant (right) cell clones expressing doxycycline-inducible MX2 in the presence (open bars) or absence (filled bars) of doxycycline (Dox) and presence or absence of CsA. Alleles of CypA in knock-out clones detailed in (B) are indicated. Titers are represented as mean + sem of infectious units (IU) per pg of reverse transcriptase (RT); left: n≥6 technical replicates combined from three independent experiments; right: n≥3 technical replicates representative of four independent experiments. Statistical analysis in S1 File. Bottom: western blot analysis of doxycycline-inducible MX2, CypA, and tubulin loading control in the indicated cell clones.
(PDF)

**S3 Fig. Validation of CA binding specificity by CypA mutants.** HIV-1 WT, N57A, or G89V CA tubes were assembled *in vitro* and incubated with lysates from control or CypA-mutant HT1080 cells as indicated. The reaction mixtures were subjected to centrifugation to separated pulled-down (bound) fractions from unbound proteins in supernatants. Lane 1: cellular lysates; Lane 2: supernatant from control experiments without CA tubes; Lanes 3–5: supernatants after incubating cellular lysates with CA tubes; Lane 6: pulled-down fraction from control experiment in the absence of CA tubes; Lanes 7–10: proteins bound to CA tubes.
(PDF)

**S4 Fig. The effects of CsA on retroviral sensitivity to MX2 are the direct result of blocking CA-CypA interactions.** Infection of control or CypA-mutant HT1080 cells (one representative clone each) expressing doxycycline-inducible MX2 in the presence (open bars) or absence (filled bars) of doxycycline (Dox) and present or absence of CsA with indicated GFP reporter viruses. Titers are represented as mean + sem of infectious units (IU) per pg of reverse transcriptase (RT), n≥6 technical replicates combined from two-three independent experiments. Statistical analysis in S1 File.
(PDF)

**S5 Fig. Effects of MX2 on retroviruses in CypA knock-out and mutant cells.** Data from Figs 1 and S4 shown as a ratio (fold change) of -Dox (-MX2)/+Dox (+MX2) in the presence (open bars) or absence (filled bars) of CsA. Average fold change calculated from three technical replicates per experiment; shown is mean + sem of log2(fold change) from two-three independent

experiments.
(PDF)

**S6 Fig. Determinants for MX2 activity in the absence of CA-CypA interactions.** A) Infectivity of GFP reporter viruses in HeLa cells expressing doxycycline-inducible C-terminally myc-tagged MX2, MX2 mutants, or MX1$_{MX2-NTD}$ in the presence (open bars) and absence (filled bars) of doxycycline (Dox). MX2 mutants are color-coded by domain/effect as in Fig 2A. Cells were infected in the presence of CsA where indicated. Titers are represented as mean + sem of infectious units (IU) per pg of reverse transcriptase (RT), n≥8 technical replicates combined from two-six independent experiments. Statistical analysis in S1 File. B) Western blot analysis of doxycycline-inducible MX2-myc and tubulin loading control.
(PDF)

**S7 Fig. Determinants for MX2 activity in the absence of CA-CypA interactions.** Data from Figs 3 and S6 shown as a ratio (fold change) of -Dox (-MX2)/+Dox (+MX2). Average fold change calculated from four technical replicates per experiment; shown is mean + sem of log2 (fold change) from two-six independent experiments.
(PDF)

**S8 Fig. Restriction of HIV-1 infection by GTPase-deficient MX2 in CypA mutant cells.** A) Infection of control and CypA-mutant HT1080 cell clones (one representative clone each) stably transduced with doxycycline-inducible myc-tagged MX2 (left) or MX2$_{T151A}$ (right) in the presence (open bars) or absence (filled bars) of doxycycline and presence or absence of CsA with the indicated GFP reporter viruses. Titers are represented as mean + sem of infectious units (IU) per pg of reverse transcriptase (RT), n≥12 technical replicates combined from five independent experiments. Statistical analysis in S1 File. B) Western blot analysis of doxycycline-inducible MX2-myc, CypA, and tubulin loading control in the indicated cell clones.
(PDF)

**S9 Fig. Effects of MX2$_{T151A}$ on retroviruses in CypA knock-out and mutant cells.** Data from S8 Fig shown as a ratio (fold change) of -Dox (-MX2)/+Dox (+MX2) in the presence (open bars) or absence (filled bars) of CsA. Average fold change calculated from three technical replicates per experiment; shown is mean + sem of log2(fold change) from five independent experiments.
(PDF)

**S10 Fig. Phosphorylation at residue T151 does not determine CypA-independent MX2 activity.** A) Infection HeLa cells stably transduced with doxycycline-inducible myc-tagged MX2 or MX2$_{T151D}$ in the presence (open bars) or absence (filled bars) of doxycycline and presence or absence of CsA with the indicated GFP reporter viruses. Titers are represented as mean + sem of infectious units (IU) per pg of reverse transcriptase (RT), n≥6 technical replicates combined from two-seven independent experiments. Statistical analysis in S1 File. B) Western blot analysis of doxycycline-inducible MX2-myc and tubulin loading control.
(PDF)

**S11 Fig. Phosphorylation at residue T151 does not determine CypA-independent MX2 activity.** Data from S8 Fig. shown as a ratio (fold change) of -Dox (-MX2)/+Dox (+MX2) in the presence (open bars) or absence (filled bars) of CsA. Average fold change calculated from three technical replicates per experiment; shown is mean + sem of log2(fold change) from five independent experiments.
(PDF)

**S12 Fig. Restriction by GTPase-deficient MX2 in the absence of CA-CypA binding is not mediated by known CA-GTPase domain interactions.** A) Infection of HeLa cells stably transduced with doxycycline-inducible myc-tagged MX2 with or without mutations in the N-terminal triple-arginine motif ($_{11}$AAA$_{13}$), T151A, or CA-binding residues in the GTPase domain (SSE) in the presence (open bars) or absence (filled bars) of doxycycline and presence or absence of CsA with the indicated GFP reporter viruses. Titers are represented as mean + sem of infectious units (IU) per pg of reverse transcriptase (RT), n = 9 technical replicates combined from three independent experiments. Statistical analysis in S1 File. B) Western blot analysis of doxycycline-inducible MX2-myc and tubulin loading control.
(PDF)

**S13 Fig. Restriction by GTPase-deficient MX2 in the absence of CA-CypA binding is not mediated by known CA-GTPase domain interactions.** Data from Figs 5 and S12 shown as a ratio (fold change) of -Dox (-MX2)/+Dox (+MX2) in the presence (open bars) or absence (filled bars) of CsA. Average fold change calculated from three technical replicates per experiment; shown is mean + sem of log2(fold change) from three-five independent experiments. $_{11}$AAA$_{13}$ = RRR$_{11-13}$AAA; SSE = G184S/N260S/Q351E.
(PDF)

**S14 Fig. Antiviral activity of GTPase-deficient chimeric MX1$_{MX2-NTD}$ proteins.** A) Infection of HeLa cells stably transduced with doxycycline-inducible myc-tagged MX1 or MX1$_{MX2-NTD}$ with the indicated mutations in the presence (open bars) or absence (filled bars) of doxycycline and presence or absence of CsA with the indicated GFP reporter viruses. Titers are represented as mean + sem of infectious units (IU) per pg of reverse transcriptase (RT), n$\geq$9 technical replicates combined from three-seven independent experiments. Statistical analysis in S1 File. B) Western blot analysis of doxycycline-inducible MX1$_{MX2-NTD}$ or MX1-myc and tubulin loading control.
(PDF)

**S15 Fig. Antiviral activity of GTPase-deficient chimeric MX1$_{MX2-NTD}$ proteins.** Data from Figs 6 and S14 shown as a ratio (fold change) of -Dox (-MX2)/+Dox (+MX2) in the presence (open bars) or absence (filled bars) of CsA. Average fold change calculated from three technical replicates per experiment; shown is mean + sem of log2(fold change) from three-seven independent experiments.
(PDF)

**S16 Fig. Experimental design for Nup/NTR knockdown.** A) Experimental strategy to investigate the roles of Nups and NTRs in antiviral activity of MX2 and MX2$_{T151A}$. For a detailed description, refer to the Materials and Methods. B) Western blot analysis of doxycycline-inducible MX2 and MX2$_{T151A}$ and tubulin loading control.
(PDF)

## Acknowledgments

We thank Dr. Kevin McCarthy for support, consultation, and assistance in generation of PyMOL images; Mariah Cashbaugh and Ananya Venbakkam for technical support. Color-blind safe "muted" qualitative color schemes from Paul Tol were used for figures [59].

## Author Contributions

**Conceptualization:** Melissa Kane.

**Data curation:** Melissa Kane.

**Formal analysis:** Melissa Kane.

**Funding acquisition:** Melissa Kane.

**Investigation:** Bailey Layish, Ram Goli, Haley Flick, Szu-Wei Huang, Robert Z. Zhang, Melissa Kane.

**Methodology:** Szu-Wei Huang, Mamuka Kvaratskhelia, Melissa Kane.

**Visualization:** Melissa Kane.

**Writing – original draft:** Melissa Kane.

**Writing – review & editing:** Robert Z. Zhang, Mamuka Kvaratskhelia, Melissa Kane.

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
