## [Decision Letter · Decision Letter 0]

15 Jan 2024

Dear Dr. Kane,

Thank you very much for submitting your manuscript "Virus specificity and nucleoporin requirements for MX2 activity are affected by GTPase function and capsid-CypA interactions" for consideration at PLOS Pathogens. As with all papers reviewed by the journal, your manuscript was reviewed by members of the editorial board and by several independent reviewers. The reviewers appreciated the attention to an important topic. Based on the reviews, we are likely to accept this manuscript for publication, providing that you modify the manuscript according to the review recommendations.

Sincerely,

Edward M Campbell, PhD

Academic Editor

PLOS Pathogens

Richard Koup

Section Editor

PLOS Pathogens

Kasturi Haldar

Editor-in-Chief

PLOS Pathogens

orcid.org/0000-0001-5065-158X

Michael Malim

Editor-in-Chief

PLOS Pathogens

orcid.org/0000-0002-7699-2064

Reviewer Comments (if any, and for reference):

Reviewer's Responses to Questions

**Part I - Summary**

Reviewer #1: In this article Layish and colleagues investigate some of the determinants governing the anti-lentiviral activity of human MX2. They have performed a large array of experiments analysing the effect of MX2 mutants on many lentiviruses and in different conditions. They have provided important pieces of evidence, nonetheless the demonstration that the effect of CSA treatment on HIV-1 infected cells is directly due to its effect on the CA-CypA interaction. They follow on showing how the regulation of MX2’s antiviral activity involves the association of CypA with the viral capsid, the GTPase activity of the protein, the presence of specific nuclear pore components on a cell line-based dependent manner. This article provides valuable information on the antiviral activity of MX2 and aids to understand the highly complex nature of its regulation by viral and cellular factors. However, I think the article is missing an overall conclusion (a punch line) on its findings. While the data clearly shows the important role of CypA and GTP binding on the regulation of MX2’s antiviral activity, with the interesting comparison of wild type and N57S viruses, it lacks a hypothesis or model that tries to tie together the findings. How do the authors think everything comes at play to regulate MX2’s activity? Is GTP binding changing how or even where MX2 interacts with CA and that makes it “CypA independent”, and down the line lead to the engagement of different nuclear pore proteins? Does MX2 has a GTPase-dependent and a GTPase-independent inhibitory mechanism?

Reviewer #2: The manuscript by Layish et al. describes a comprehensive study to determine the specificity of Mx2 restriction in HT1080 and HeLa cells to different viral isolates and by modification of endogenous CypA by Knock-In by Cas9 and/or CsA presence.

The manuscript answers many unknowns in the topic and is very comprehensive and complete in the approach. In general the results support the conclusions of the discussion. The description of GTPase deficient Mx2 mutants together with measurement of CA determinants, CypA/CsA activity further sheds light on how Mx2 targets incoming HIV-1 and articulates with other cell proteins.

Some of the discussion should accommodate other possibilities on indirect effects on Mx2 restriction by CypA capability to isomerize other host proteins as the only direct binding data shown in the article is the nanotube data in Fig S3 (It is correctly stated in results lines 150 to 153, but not in the discussion). The relationship with TRIM5alpha should also be discussed.

There are also several items that should be improved or corrected as the data is extensive making it certainly hard to follow to a broader audience.

**Part II – Major Issues: Key Experiments Required for Acceptance**

Reviewer #1: (No Response)

Reviewer #2: Line 115: HIV-1WT infection was potently inhibited in TRIM-huCypA(R69H) and not D66N and D66N/R69H. I believe this to be an edditing error as the conclusions are in accordance to the data in Fig 1 and Fig S1.

Line 122 (and others): specify and clarify on when and how many clones were used in the experiments, it is not clear if the data in the figures are a mix of the several described clones in FigS2, only the best one or if there is divergence in their phenotypes.

Fig 7: The results of Nup85, Sec13 and other depletions to Mx2 restriction are not described. The authors should discuss and hypothesize why MX2 does not resctrict upon depletion but but MX2 T151A does (for HIV-1 WT).

-All data is normalized to IU/pgRT, as changing HIV CA results in very dramatic changes in infectivity, it is possible that some results are more susceptible to saturation than others. Authors should discuss this possibility and address it.

**Part III – Minor Issues: Editorial and Data Presentation Modifications**

Reviewer #1: 1. Lines 114-116 say “As expected, HIV-1WT infection was potently inhibited (~200-fold) by TRIM-huCypA D66N and TRIM-huCypA D66N/R69H in a manner reversible by CsA treatment, while TRIM-huCypA R69H was inactive against HIV-1WT”. I think the authors mean quite the opposite, since mutant R69H is the only one inhibiting HIV-1 infection, while those having the D66N mutation do not.

2. Lines 375-377: describes the construction of the different omkTRIM-CypA plasmids as having “the N-terminal domains (RIGN, B-Box and coiled coil) from owl monkey (omk) TRIMCyp followed by an HA tag and cloning sites (NotI/SalI) that allowed introduction of various proteins”. As it is stated one would think that the constructs go as TRIM-HA-CypA, while they are HA-TRIM-CypA. Therefore, I believe it is best to say that the HA tag is followed by the N-terminal domains from omkTRIMCyp.

3. Lines 133-135: I find the wording a bit confusing. My first impression was that CSA treatment reduced levels of infectivity and a lack of MX2 sensitivity in CypA-/-, CypAD66N, and CypAD66N/R69H. I now know that what they point is that, in the absence of CSA, those CypA proteins reduced levels of infectivity and a lack of MX2 sensitivity, as addition of CSA does for wt CypA. I think rewriting this sentence to make it clearer would improve the manuscript.

4. Line 177: an “in” is missing between CSA and HT1080. Also, in Fig S5 (HeLa cells) the antiviral activity of MX1(NTD_MX2) is very much cancelled by CSA.

5. Line 185: I think the sentence GTP-binding deficient mutant K131A makes clearer this mutant does not bind GTP.

6. Lines 186-187: I fail to understand the K131A data. It seems to be a slight reduction on infection in Fig 3. In fact, if I am understanding the S1 statistical data correctly, this reduction is of 2.605, and significant. Therefore, K131A does inhibit HIV-1 infection in HT1080 cells, and therefore the sentence in the following lines about GTP binding being cell-type dependent wouldn’t make sense. This issue arises again in lines 336-337.

7. In Fig 3A, the effect of CSA with MX1 (NTD_MX2) is quite remarkable, the increase in HIV-1 infectivity seems to be ~3x, higher than any other protein tested. Do the authors think this is due to some MX1 determinant absent in MX2?

8. I am intrigue by the immunoblot data in Fig S6. In CypA D66N cells the level of MX2 T151A is drastically reduced compared with the other conditions, and levels of MX2 are also diminished, with tubulin levels unaffected. One could think that somehow CypA D66N is producing a drop on the amount of cellular MX2, but this trend is not seemed in Fig 1, could the authors explain this?

9. What is the difference between the antiviral activity of T151A and T151D? It seems that both mutants have the same inhibitory profile.

10. In Fig 5, in the first lane under the bar chart where the presence or not of mutations in different MX2 positions are depicted, an indication of the amino-terminal domain is missing.

11. I find Fig 5A too difficult to follow. The combination of 8 different conditions on a single panel, while excellent for saving space, make it too complicated to comprehend. I would suggest the authors to split them in 2 or 3 panels, maybe based on the type of mutation (i.e., NTD, T151 and G domain).

12. Fig 6 is rather difficult to follow, especially because the most significant comparison of MX1(NTD_MX2) mutants is with the corresponding MX2 mutants, rather than MX1. Therefore, having MX1(NTD_MX2) mutants side to side with MX2 in Fig 6 would be more informative.

13. Also in Fig 6, how are MX1 T103A/D mutants inhibiting N57S in the presence of CSA?

14. Fig 7: indentation of B-D) needs correction.

15. It would be interesting to see the effect of T151A on HIV-1 T210K mutant viruses, since this residue is involved in the binding of MX2’s NTD and could give some more information about the viral determinants at play on the particular capabilities of T151A

16. Would the NUPs requirement for MX2 in CypA KO or D66N mutant cells be similar to that of T151A?

Reviewer #2: Line 163: the authors claim that inhibition of other primate lentiviruses by MX2 is independent of CA-CypA interactions. Since not all other primate lentiviruses were tested, the authors should specify which ones are representing such statement.

-Usage of HIV-1NL4-3 for WT G89V A92E and G94D and NHGCapNM for N57S, N57D, N74D, G208R and T210K should be justified. What other differences are there in the sequences between NL4-3 and NHGCapNM, if they exist what could be a confounding factor in the results?

-TRIM5alpha restriction upon blocking CypA interaction to the capsid is mentioned in the introduction but not discussed as a possible factor into many of the shown phenotypes. How do the authors see that Mx2 presence removed TRIM5alpha restriction by the proteosome (+CsA)?

Improvement of figures:

-There is quite a lot of data, and it is an impressive body of work. However, the authors should consider another representation (in parallel of as supplementary) that would be easier to read the restriction/enhancement such as ratios, log2FC or alike.

-Some graphs compare 0.1 to 100, other 0.001 to 10 IU/pg RT, many of the restriction in the higher IU/pg RT look smaller than they are, for example in Line 136 there's a "slight enhancement of infection" by G89V that appears minimal as the scale compressed the important information. For example Fig S4 N74D Ctrl data 2x~5x ratios have minimal differences in the graph and are hard to interpret.

-Fig 2/3 color codes appear to be "by motif", please make it clear in the describing text or as a figure legend.

-Fig 5. 11RRR15 is missing in the figure (above green T151).

-Fig 6 and Fig S9 should have WT in legend for clarity.

-Some Circles in Fig 7 only show one circle (-Dox), certainly is because they overlap, but the authors should find a way to make it clear (slight shift in the horizontal axis for example). An example is NUP153KD in HIV-1WT-MX2.

-The statistical data is provided in supplemental data. Some of it should be clearly reported in the figures themselves (at least the ones described in results/discussion).

Text improvement comments:

Line 67: "alters interactions" should be specific as to what is the phenotype of the phosphorylation of Serins.

Line 69-70: be more specific (at least exemplify) on what are the phenotypes to alterations of residues throughout all domains of MX2.

Line 79-80: regarding enzymatic activity of CypA need references.

Line 109: The results section needs improvement on the experimental details (that are provided in methods) on whether TRIMCyp proteins are overexpressed/KI and that the TRIM-huCypA carry the TRIM "body" of owlTRIM and not human TRIM5alpha.

Line 133: Csa/HT1080 phenotype described should have a citation in this section of the text.

Line 142: It is not clear if "this mutant" is relative to N57D or N57A.

Line 191: Please specify which MX2 mutants the authors are referring to.

Line 247: Please clarify in the text to which the mutations positions "belong" to, either Mx1 or Mx2. For example K131A (K83A) give no information to what protein each belongs to respectively.

PLOS authors have the option to publish the peer review history of their article (what does this mean?). If published, this will include your full peer review and any attached files.

Reviewer #1: **Yes: **Gilberto Betancor

Reviewer #2: No

Figure Files:

Data Requirements:

Reproducibility:

References:

---

## [Editor Report · Decision Letter 1]

29 Feb 2024

Dear Dr. Kane,

We are pleased to inform you that your manuscript 'Virus specificity and nucleoporin requirements for MX2 activity are affected by GTPase function and capsid-CypA interactions' has been provisionally accepted for publication in PLOS Pathogens.

Best regards,

Edward M Campbell, PhD

Academic Editor

PLOS Pathogens

Richard Koup

Section Editor

PLOS Pathogens

Michael Malim

Editor-in-Chief

PLOS Pathogens

orcid.org/0000-0002-7699-2064
---

## [Editor Report · Acceptance letter]

18 Mar 2024

Dear Dr. Kane,

We are delighted to inform you that your manuscript, "Virus specificity and nucleoporin requirements for MX2 activity are affected by GTPase function and capsid-CypA interactions," has been formally accepted for publication in PLOS Pathogens.

Best regards,

Michael Malim

Editor-in-Chief

PLOS Pathogens

orcid.org/0000-0002-7699-2064